# Associations of Lipophilic Micronutrients with Physical and Cognitive Fitness in Persons with Mild Cognitive Impairment

**DOI:** 10.3390/nu11040902

**Published:** 2019-04-22

**Authors:** Perihan Gerger, Roopa Kalsank Pai, Tim Stuckenschneider, Julia Falkenreck, Hannah Weigert, Wilhelm Stahl, Bernd Weber, Gereon Nelles, Liana Spazzafumo, Stefan Schneider, M. Cristina Polidori

**Affiliations:** 1Ageing Clinical Research, Department II of Internal Medicine and Center for Molecular Medicine Cologne, University of Cologne, Faculty of Medicine and University Hospital Cologne, 50931 Cologne, Germany; dr.perihan.gerger@gmail.com (P.G.); julia.falkenreck@web.de (J.F.); hannah.weigert@gmail.com (H.W.); 2Institute of Experimental Epileptology and Cognition Research, University Hospital Bonn, Germany and Center for Economics and Neuroscience, University of Bonn, 53127 Bonn, Germany; roopakalsankpai@gmail.com (R.K.P.); bernd.weber@ukbonn.de (B.W.); 3Institute of Movement and Neurosciences, German Sport University, 50933 Cologne, Germany; t.stuckenschneider@dshs-koeln.de (T.S.); Schneider@dshs-koeln.de (S.S.); 4VasoActive Research Group, School of Health and Sport Sciences, University of the Sunshine Coast, Maroochydore, QLD 4558, Australia; 5Institute of Biochemistry and Molecular Biology I, Heinrich-Heine University Düsseldorf, 40204 Düsseldorf, Germany; wilhelm.stahl@uni-duesseldorf.de; 6NeuromedCampus Hohenlind, 50935 Cologne, Germany; gereon.nelles@uni-due.de; 7Epidemiologic Observatory, Regional Health Agency, I-64125 Ancona, Italy; liana.spazzafumo@regione.marche.it

**Keywords:** cognitive performance, physical activity, micronutrients, carotenoids, mild cognitive impairment, nutrition, neuropsychological tests

## Abstract

Age-associated cognitive impairment in general and dementia in particular are a global concern. Preventive lifestyle strategies are highly used but there is a lack of information on the reciprocal relationships between nutrition biomarkers and measures of both cognitive and physical performance. To fill this gap of knowledge, the relationship between plasma levels of the robust nutrition- and antioxidant defense-related biomarkers carotenoid and tocopherols and both indicators of cognitive and physical performance was investigated in a group of persons with mild cognitive impairment participating in the NeuroExercise Study at the German Sport University in Cologne, Germany. In 56 participants with full dataset, significant correlations independently of fruit and vegetable intake were found between plasma levels of β-cryptoxanthin and Timed Up&Go test (*p* < 0.05), γ-tocopherol and number of daily steps (*p* < 0.01), as well as between four out of six measured carotenoids—lutein; zeaxanthin; β-cryptoxanthin and β-carotene—and the computerized CogState International Shopping List subtest (*p* < 0.01). In light of the increasing attention towards the nutritional cognitive neuroscience of carotenoids, computerized measures of cognitive performance might be further implemented in future studies investigating the effects of lifestyle interventions against cognitive and physical impairment.

## 1. Introduction 

Socioeconomic challenges associated with the ageing of the population demand concerted response worldwide as the increasing proportion of older people leads to changing needs in health and social care, with multi- and interdisciplinarity, networking and translational approaches moving to the forefront. Among the biggest medical challenges caused by chronic conditions in advanced age, cognitive impairment plays a major role [1,2]. Cognitive decline displays as diminished orientation or ability to remember, judge, understand and reason. It may be accompanied by a decrease in several other cognitive abilities and is extremely frequent in advanced age. For severe cognitive impairment, the term dementia (from the latin *de*, “out of”, and *mens*, “mind”) is commonly used. Cognitive decline and cognitive impairment with or without dementia are multifactorial syndromes rather than a single *one cause–one mechanism* disease.

To the main features of cognitive decline with and without dementia—multifactoriality, heterogeneity, poor diagnosis, increasing prevalence [1,2,3]—one adds causing major public health concern, i.e., that dementia is not curable. Therefore, attention is shifted towards the need of early diagnosing cognitive changes to slow down their progression. Age-related cognitive decline, subjective cognitive impairment (SCI) [4], and mild cognitive impairment (MCI) [5] are therefore object of a large body of investigations prompted at identifying the best possible preventive strategies. While the frantic, challenged search for effective antidementia drugs is ongoing, preliminary studies on the role of vascular- and lifestyle-related preventive strategies show that vascular risk control and lifestyle improvement are indeed able to slow down the progression of cognitive impairment [2,6,7]. Among lifestyle interventions, cognitive training programs, physical exercise interventions, and dietary strategies gained a great deal of attention recently (see present special issue). Several of these studies are based upon the evidence that oxidative stress, a critical pathophysiological mechanism in the onset and progression of cognitive impairment [8], can be substantially influenced by physical activity [9] and nutrition [10]. In particular, several biomarkers of oxidative stress and indicators of antioxidant micronutrient defense against free radicals have been shown to be associated with cognitive impairment with and without dementia [11]. Although the results of these studies are highly promising, the interactions between the different components of lifestyle across the course of cognitive impairment have not been clearly identified yet. This might be relevant to explain the conflicting results of lifestyle interventions including the multidomain ones [12] and to plan more personalized treatments in the future.

To fill this gap of knowledge, we investigated the relationship between robust nutrition- and antioxidant defense-related biomarkers including six carotenoids, two tocopherols and retinol, and indicators of both cognitive and physical performance in a group of persons with MCI participating in the NeuroExercise Study at the German Sport University in Cologne, Germany.

## 2. Participants and Methods

### 2.1. Participants

All participants were recruited through the NeuroExercise Project [13,14], a multi-centered randomized controlled trial of exercise therapy in persons with MCI according to Albert et al. [15] across three European countries [13]. For the purpose of the present sub-study, the participants were recruited in Germany at the German Sport University (GSU). The study was conducted in accordance with the declaration of Helsinki (1975) and approved by the research ethics committee of the GSU. Participants were recruited through newspaper advertisements and all of them provided informed consent to the study procedures. Participants were included if the Montreal Cognitive Assessment (MoCA) [16] scored ranged between 18 and 26; if they had a stable medical condition for more than 6 months and stable medication for more than 3 months; adequate visual and auditory acuity to complete neuropsychological testing, electrocardiogram without significant abnormalities that might interfere with the study; physical ability sufficient to allow performance of endurance exercise training; capacity to provide written and dated informed consent form, as well as having complete physical examination including a symptom-limited cardiopulmonary exercise test. To distinguish between amnestic and non-amnestic MCI, agreed education adjusted cut-offs of −2 Standard Deviation (SD) for low education (<10 years of education), −1.5 SD for the middle group (10–13 years of education), and −1 SD for the highly educated (>13 years of education) were derived from the delayed recall portion of the age-adjusted delayed memory index of the Repeatable Battery for the Assessment of Neuropsychological Status (RBANS) [17] (Score of < 85) as previously described [13,14].

Exclusion criteria were diagnosis of Alzheimer’s disease (AD) or other type of dementia, history of familial early-onset dementia; enrollment in any investigational drug study; history in the past 2 years of epileptic seizures; any major psychiatric disorder (a clinical diagnosis of major depressive disorder, bipolar or schizophrenia); past history or MRI evidence of brain damage, including significant trauma, stroke, hydrocephalus, mental retardation, or serious neurological disorder as well as carotid stent or severe stenosis; history of myocardial infarction within previous year, congestive heart failure (New York Heart Association Class II, III or IV); uncontrolled hypertension or hypotension (systolic blood pressure >200 mm Hg and/or diastolic blood pressure >110 mm Hg at rest); unstable cardiac, renal, lung, liver, or other severe chronic disease; type 2 diabetes mellitus with hypoglycemia in the last 3 months; significant history of alcoholism or drug abuse within last 10 years; engagement in moderate-intensity aerobic exercise training for more than 30 min, 3 times per week, during past 2 years; history of vitamin B12 deficiency or hypothyroidism (stable treatment for at least 3 months is allowed); and serious or non-healing wound, ulcer, or bone fracture [13].

After signing informed consent and passing the eligibility criteria, all participants underwent baseline assessment, including collection of history and lifestyle habits, neuropsychological testing, testing of general physical and cardiovascular performance. Participants were then randomly allocated to one of the three interventions (aerobic vs. stretching and toning 3 × 45 min exercise sessions per week over 12 months vs. usual care without exercise counseling) as described before [13,14].

For the purpose of the present study baseline results are reported on participants with complete sets of neuropsychological assessments (MoCA, CogState, TMT A and B, letter and category fluency tests), physical tests (LASA Physical Activity Questionnaire, LAPAQ, and Timed Up and Go, TUG) and lipophilic micronutrient plasma levels (retinol, six carotenoids and two tocopherols) as described below.

### 2.2. Neuropsychology

The MoCA, used as a broad measure of global cognitive function, is a one-page 30-point test administered in 10 min which consists of 13 tasks covering the following eight cognitive domains: visuospatial/executive functions, naming, verbal memory registration and learning, attention, abstraction, delayed verbal memory, and orientation. It has demonstrated high sensitivity and specificity as a cognitive screening instrument and has been validated to detect MCI [16]. Cognitive performance was assessed by a gamified computerized neuropsychological test battery measuring six cognitive domains. The test battery consisted of a computer-based CogState Battery including the International Shopping List Task (ISLT)—immediate and delayed recall, Detection Task, Identification Task, One Back Task (ONB), and One Card Learning Task (https://cogstate.com/) [18], verbal fluency [19,20] and Trail Making Test (TMT) [21].

*Verbal memory* was assessed by ISLT, *psychomotor function* by the Detection Task, *executive function* by TMT-B, Letter Fluency and Category Fluency. *Attention* was assessed by the Identification Task and TMT-A. *Working memory* was measured by One Back Task, and *Visual memory* by the One Card Learning Task. The ISLT is a 12-word, four-trial tests, where the total number of correct responses made in remembering the list on three consecutive trials at a single session and after a delay is recorded. The ISLT has been shown to have good sensitivity to verbal memory impairment [18]. The Detection Task measures psychomotor functioning and speed of processing. Participants must respond as quickly as possible, by pressing a keyboard button, when a playing card displayed on the computer screen shown face down flips over. Reaction time is measured with lower scores indicating better performance. The Identification Task measures visual attention. Participants must decide whether a playing card presented on screen is red, by pressing the ‘Yes’ or ‘No’ button. Reaction time is measured and lower scores indicate better performance. The One Back Task assesses working memory. Participants are presented with a sequence of playing cards in the center of the screen and must decide if the card presented is the same as the one shown immediately before. The One Card Learning Task measures visual learning and memory. Participants are presented with a succession of playing card on screen, and must decide if the card currently displayed has been displayed previously. Accuracy of performance is measured, with higher scores indicating better performance. A number of studies have found that the CogState battery of tests are sensitive to detecting cognitive impairment in mild to moderate AD and amnestic MCI populations relative to healthy matched controls [18]. 

Verbal fluency was assessed by Letter Fluency [19] and Category Fluency [20]. For the Letter Fluency test participants were asked to generate in one minute as many words as possible beginning with a specific initial. This task was repeated three times with three different letters (e.g., L, B, S). For the category fluency test, participants must give as many examples of animals as possible within one minute. TMT [21] was completed as a paper-and-pencil-based task. The TMT consists of two sub-trials. TMT-A require individuals to sequentially connect 25 encircled numbers on a sheet of paper, while TMT-B require participants to draw a line, alternating between numbers and letters in ascending order.

### 2.3. Physical Activity Assessments

Physical activity and mobility were assessed through the Timed Up and Go (TUG) test [22] as well as through the self-reported LASA Physical Activity Questionnaire, which is a valid and reliable interview-administered questionnaire able to captures physical activity across six categories (walking outdoors, bicycling, gardening, light household activities, heavy household activities, and sport and exercise activities) over the preceding 14 days [23]. Mean daily activity scores, and mean time spent in sport and exercise activities were calculated by summing the reported activities in minutes and dividing those by the number of days.

### 2.4. Nutritional Analyses

For the nutrient analysis, the fruit and vegetable intake of each subject was calculated in grams. Subjects filled out a paper-based food frequency questionnaire (“Ernährungsfragebogen”) used in the German Health Interview and Examination Survey for Adults (DEGS, “Studie zur Gesundheit Erwachsener in Deutschland”) by the Robert Koch Institute. This questionnaire consists of 57 questions. Fifty-three questions are concerning the frequency of consumption of individual food items and the usual portion size when consumed; the remaining questions are to provide further detail on dietary patterns, such as the different types of fats/oils used and whether subjects do not eat certain foods. For each of the 53 questions concerning a food item, there are up to 3 sub-parts. Subjects are first asked how often they have consumed this item in the last 4 weeks (28 days). If they have consumed it at all, they are asked to indicate the usual portion size of the food item per intake. For some questions, they are asked a further qualitative question about the food item. Furthermore, for each question, the questionnaire provides a representative image of portion size for that food. The calculation of fruit and vegetable intake was as follows. The calculation for fruits took into account fresh fruit (e.g., apples, bananas) and preserved fruit (e.g., compote, canned fruit); the calculation for vegetables took into account fresh vegetables (e.g., lettuce, salads), cooked vegetables and pulses (e.g., beans, peas, lentils). After the exclusion process was complete, the amount of each food item consumed was calculated for the remaining subjects by multiplying the frequency of consumption with the portion size indicated in the questionnaire. Portion sizes were converted to gram amounts according to the reference values used in DEGS1 [24]. The gram amounts for the food items that made up the food groups “fruits” and “vegetables” respectively were then added together. For one subject, all three of the items that made up the group “vegetables” were missing values, and so this subject’s vegetable intake was considered as a missing value.

For the measurement of lipophilic antioxidant micronutrients, blood was collected in a heparinized tube and immediately centrifuged. Plasma was stored frozen at −80 °C until analysis, which was performed as described before [25]. Briefly, carotenoids including lutein, zeaxanthin, β-cryptoxanthin, lycopene, and α- and β-carotene were analyzed by HPLC with UV-vis detection at 450 nm according to Stahl et al. [26]. A second UV-vis detector was connected in series and set at 325 and 292 nm for quantitation of retinol (vitamin A), and α- and γ-tocopherol (vitamin E), respectively. Recovery from the column was 90% for each micronutrient. The calibration curves were linear from 0 to 1000 nmol/L for all carotenoids, with correlation coefficients 0·99. The intra- and inter-assay precision varied between 5 and 15%.

### 2.5. Statistics

For the statistical analysis, continuous variables are presented as mean ± standard deviation (SD), categorical variables as count and percentage. Correlations between plasma concentrations of micronutrients (lutein, zeaxanthin, β-cryptoxanthin, lycoepene, α-carotene, β-carotene, α-tocopherol, γ-tocopherol and retinol) and measures of cognitive (MoCA, CogState, TMT A and B, letter and category fluency tests) and physical performance (LASA Physical Activity Questionnaire, LAPAQ, and Timed Up and Go, TUG) were calculated using Pearson’s correlation coefficient or partial correlation coefficient (r) controlled for fruit/vegetables intake. Statistical significance was defined as a two-tailed *p* value < 0.05. Data analysis was carried out with the SPSS/Win program version 23.0 (SPSS, Chicago, IL, USA).

## 3. Results

Of the 121 NeuroExercise study participants included at the GSU, 56 had the full dataset including neuropsychological assessment, physical fitness analysis, as well as plasma levels of micronutrients including retinol, six carotenoids, and two tocopherols. Demographic, neuropsychological, and physical characteristics of the study participants as well as their laboratory values are displayed in Table 1 and are in agreement with observations described in the literature [27,28].

The results of the analysis of the relationship between measures of cognitive (MoCA, TMT A, TMT B, Letter Fluency, Category Fluency and CogState) and physical (Activity Monitor, Number of Steps, LAPAQ, TUG) performance are displayed in Table 2.

When analyzing the associations between laboratory parameters and cognitive/physical measures, significant correlations independently of fruit and vegetable intake were found between plasma levels of β-cryptoxanthin and TUG (*p* < 0.05), γ-tocopherol and number of daily steps (*p*< 0.01), as well as four out of six measured carotenoids—lutein, zeaxanthin, β-cryptoxanthin, and β-carotene with ISLT (*p* < 0.01). (Table 3).

Finally, we were able to identify two groups of individuals based upon micronutrient levels. The first group (7 subjects) was called “supermicro” as persons displayed all micronutrient levels above the median value (the Boolean AND operation was used in the selection algorithm). These were compared to the group of the remaining 49 participants. This comparison underlined the supermicro group as indeed better both physically (Timed Up&Go test, 8.0 ± 0.9 s vs. 9.2 ± 1.4 s, *p* = 0.002) and cognitively (One card learning subtest of the CogState, 58.6 ± 8.3 vs. 65.7 ± 8.4, *p* = 0.04) performing than the rest of the participants.

## 4. Discussion

The main result of the present study is that plasma levels of several lipophilic antioxidant micronutrients are significantly associated, independently of fruit and vegetable intake, with validated, accurate measures of both cognitive and physical performance in persons with MCI. To our knowledge, no studies have so far explored a broad spectrum of the most robust biomarkers of nutrition and antioxidant defense in clinically highly characterized patients comprehensively assessed as far as both cognitive and physical performance are concerned. In particular, the use of a large battery of neuropsychological tests including a validated, accurate gamified computerized testing with high clinimetric properties (CogState) allows a detailed description of the association between selected micronutrients and specific cognitive abilities. This represents a further step in the field of nutritional cognitive neuroscience [29]. Carotenoids are robust biomarkers of dietary exposure which have been previously shown to be associated with global measures of cognition [27,30] and lipid profile in patients with cognitive impairment with and without dementia [28,30,31,32]. In addition, selected tocopherols and carotenoids have been shown to be directly associated with fruit and vegetable intake [28,33] as well as positively associated with cognitive performance and inversely associated with markers of oxidative stress in healthy subjects independent of age, gender, and fruit/vegetable intake, suggesting their protective role even in the absence of disease [27,28]. In the present study, plasma concentrations of several carotenoids were strongly correlated with the ISL task of the CogState battery, a valid gamified computerized testing method of which the ISL sensitively and reliably measures verbal learning and memory [18]. Interestingly, a significant association was previously found in persons with subjective cognitive impairment (SCI) between the ISLT and the endothelial peripheral arterial tonometry index (EndoPAT Index), a measure of endothelial function [34], considered an important mediator of cognitive impairment [8]. In addition, it should be considered that the present observations are in strong agreement with previous reports of a relationship between higher plasma concentrations of lipophilic micronutrients and specific high cortical functions, in particular immediate memory and measures of global cognition [28,30,31,32,33,34]. The association of plasma concentrations of circulating protective micronutrients—no matter whether as markers of nutritional exposure or pure indicators of defense against redox imbalance and oxidative stress (Table 3)—with both physical and cognitive performance reflect the tight relationship between these two families of functions, and has recently been the object of great attention and also confirmed in the present investigation (Table 2).

Interestingly enough, among 331 candidate (bio)markers investigated in the MARK-Age study, lower levels of β-cryptoxanthin and zeaxanthin were found, among 2220 randomly recruited age-stratified persons, in those who were physically, cognitively, or psychologically frail [35]. In this study, levels of β-cryptoxanthin and zeaxanthin were inversely associated with risk of being cognitively frail after adjusting for confounders. In our study, plasma levels of β-cryptoxanthin were also correlated with the TUG test, a marker of balance and increased fall risk and of physical frailty. Carotenoids may indicate the involvement of oxidative stress and inflammation in frailty, both physical and cognitive.

It is important to briefly point on the biochemistry of carotenoids to better understand their special role in nutritional cognitive neuroscience. As natural pigments present in plants, animals, and microorganisms, carotenoids reduce reactive byproducts such as reactive oxygen species (ROS) thereby acting as antioxidants. ROS, like other free radicals, are known to be potent mediators of neurodegeneration [8]. Carotenoids’ polarity - xanthophylls such as astaxanthin, β-cryptoxanthin, lutein, and zeaxanthin are polar while carotenes such as α-carotene, β-carotene, and lycopene are nonpolar- determines their differential distribution in the human body, with xanthophylls accounting for 66–77% of the total carotenoids in the frontal and occipital lobes of the human brain [30]. In addition, lutein and zeaxanthin are the only two carotenoids that cross the blood–retina barrier to form macular pigment in the eye [36] and lutein is the dominant carotenoid in human brain tissue [37,38,39]. These biochemical characteristics strongly indicate that these micronutrients are determinant for nervous tissue physiology and functioning. Furthermore, lutein is the major carotenoid in brain tissue despite not being the major carotenoid in matched serum (an indicator of dietary intake), which implies a preferential uptake into brain tissue [39]. Lutein and zeaxanthin in macula were found to be significantly correlated with their levels in matched brain tissue in primates [40], suggesting that macular pigment can be used as a biomarker in brain tissue. This is of interest, given that a significant correlation was found between macular pigment density and global cognitive function in healthy older adults [41,42]. In brain tissue of decedents from a population-based study, lutein was found to be consistently associated with a wide range of cognitive measures [43] which, similar to our study, included executive functions, language, learning, and memory. These functions were in turn found to be specifically associated to carotenoid contents in specific brain regions. 

The main limitation of our study is the low sample size. However, the observed values of micronutrient status, cognitive and physical ability are in agreement with those present in the literature. In addition, the accurate inclusion and exclusion criteria as well as the very comprehensive neuropsychological battery and the use of the most robust biomarkers of dietary exposure measured through high performance analytics guarantee the interpretability of the results. Another issue might be that, although the evidence for a role of carotenoids in cognitive function is accumulating, the studies discussed thus far, including the present one, are correlative and do not demonstrate cause and effect. However, in a double-blinded, placebo-controlled trial of women who received lutein supplementation (12 mg/d), docosahexaenoic acid supplementation (800 mg/d), or a combination of the two for 4 months, verbal fluency scores improved significantly in all three treatment groups. Memory scores and rates of learning improved significantly in the combined treatment group, who also displayed a trend toward more efficient learning [43]. Taken together, these observations suggest that at least lutein could influence cognitive function.

In conclusion, plasma levels of carotenoids were shown to be significantly associated with indices of both physical and cognitive frailty in the present study. In light of the increasing attention towards the nutritional cognitive neuroscience of carotenoids [44,45] as well as the vascular component of nutritional cognitive neuroscience [6,7,8,10,11,27,28,29,30,31,32,33,34,46], the use of computerized measures of cognitive performance might be further implemented in future studies investigating the effects of lifestyle interventions against cognitive and physical impairment. 

## Figures and Tables

**Table 1 nutrients-11-00902-t001:** Demographic and clinical characteristics and laboratory values of the study participants.

Parameters	Values
Age (years)	73.1 ± 5.8
Gender [*n* (%) Female]	26 (46)
BMI	25.8 ± 3.4
Fruit and vegetable intake (g)	466 ± 371
LAPAQ (min)	229.8 ± 144.7
Steps/day	9895.7 ± 3714.7
TUG (s)	9.1 ± 1.5
MoCA (score 0/30)	23.2 ± 2.1
TMT A (s)	50.4 ± 20.6
TMT B (s)	145.4 ± 72.1
Letter Fluency (number of words/min)	11.6 ± 3.8
Category Fluency (number of words/min)	17.2 ± 4.6
DET (ms log10)	2.63 ± 0.09
IDN (ms log10)	2.80 ± 0.07
OCL (number of correct inputs)	64.8 ± 8.6
ONB (number of correct inputs)	89.6 ± 10.03
ISLT (number of recalled items)	20.6 ± 5.1
Lutein (µM)	0.46 ± 0.33
Zeaxanthin (µM)	0.08 ± 0.11
β-Cryptoxanthin (µM)	0.41 ± 0.47
Lycopene (µM)	0.51 ± 0.33
α-Carotene (µM)	0.10 ± 0.09
β-Carotene (µM)	0.74 ± 0.65
α-Tocopherol (µM)	28.9 ± 7.8
γ-Tocopherol (µM)	2.37 ± 0.84
Retinol (µM)	1.45 ± 0.42

**Table 2 nutrients-11-00902-t002:** Significant correlations between measures of cognitive and physical performance.

	MoCA	TMT A	TMT B	LAPAQ	Steps	TUG	Letter Fluency	Category Fluency	CogState
MoCA		−0.6	−0.5		0.4	−0.3	0.35	0.5	ISLT: 0.6
		<0.0001	<0.0001		0.003	0.005	0.001	<0.0001	<0.0001
TMT A			0.7		−0.3	0.3	−0.24	−0.46	IDNlog: 0.46
			<0.0001		0.03	0.02	0.03	<0.0001	<0.0001
									OCL: −0.3
									0.009
									ONB: −0.3
									0.01
									ISLT: −0.4
									0.001
TMT B						0.4	−0.34	−0.45	DETlog:
						<0.0001	0.002	<0.0001	IDNlog: −0.42
									<0.0001
									OCL: −0.37
									0.001
									ONB: −0.38
									0.001
									ISLT: −0.38
									0.001
LAPAQ									
Steps								0.34	ISLT: 0.26
								0.009	<0.05
TUG							−0.34		ISLT: −0.3
							0.004		0.009
Letter Fluency								0.48	ISLT: 0.3
								<0.0001	0.009
Category Fluency					0.34				ISLT: 0.5
					0.009				<0.0001

MoCA: Montreal Cognitive Assessment; TMT A and B: Trail Making Test A and B; TUG: Timed Up&Go.

**Table 3 nutrients-11-00902-t003:** Correlations between lipophilic micronutrients and physical/cognitive measures of the study participants. Adjusted for fruit and vegetable intake.

	r Value	*p* Value
β-Cryptoxanthin/TUG	−0.564	<0.05
γ-Tocopherol/Steps/day	0.525	<0.01
Lutein/ISLT	0.686	<0.01
Zeaxanthin/ISLT	0.723	<0.01
β-Cryptoxanthin/ISLT	0.603	<0.01
β-Carotene/ISLT	0.660	<0.01

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
