# Peer review of "Associations of Lipophilic Micronutrients with Physical and Cognitive Fitness in Persons with Mild Cognitive Impairment"

_nutrients, 2019, doi:10.3390/nu11040902_

Round 1

Reviewer 1 Report

The manuscript entitled “Associations of lipophilic micronutrients with physical and cognitive fitness in persons with mild cognitive impairment” deals with an important issue in today’s society. The most robust aspect of the present manuscript consists in the use of a comprehensive battery of test to evaluate different cognitive abilities in the participants and trying to correlate with plasma levels of different diet antioxidants, and with the physical performance analysed by different scores. The weakness consists in the low number of participants that contribute to obtain correlation with few cognitive test (but despite the low sample number, some correlations are obtained). An additional aspect has to do with the analysis of the obtained data that could probably be processed more to try to find additional correlations; not with the raw data, but by groups of patients within certain ranges in each parameter analyzed; or with the three intervention groups with respect to physical exercise, for example.

Comments

Authors indicate that participants were randomly allocated in three different interventions regarding to physical activity. Author should group these participants and check correlations with LAPAQ or TUG values, with cognitive tests results (and with MCI), and even regarding to other parameters analysed.

Author should discuss the reason why the correlation between one lipophilic micronutrient is only with one physical activity parameter or cognitive test parameter and not with the other ones. Is there correlation between the different physical activity parameters analysed? or is there correlation between the different cognitive test analysed? If there is not correlation, author should discuss the reason. If there is correlation, author should group all these activity parameters or all these cognitive test results and correlate them with the micronutrients levels. Additionally, is there correlation between fruit and vegetables intake and micronutrients levels? Authors should comment these correlations indicating even if they do not correlate with a brief discussion about why do not correlate.

Table 1 should appear in only one page (authors can make it smaller if it not fit). Some legend would be necessary to explain gender parameter and an explanation how to obtain BMI. There are some parameters that present a great error… could it be better to show SEM instead of SD? In the case of parameters with big error, some descriptive analysis of the parameter would be needed (how is the distribution? It is possible to make different groups? And do they correlate with micronutrients levels?). Some description from data obtained is needed.

What is the meaning of arguing about the levels of lutein and zeaxanthin in macula? This part of the discussion could be eliminated. It would be preferable to broaden the discussion regarding the cognitive functions that micronutrients levels correlate mainly and why other cognitive functions do not correlate.

A more complet analysis of data should be needed do to the low number of participants included.

Author Response

Reviewer 1

The manuscript entitled “Associations of lipophilic micronutrients with physical and cognitive fitness in persons with mild cognitive impairment” deals with an important issue in today’s society. The most robust aspect of the present manuscript consists in the use of a comprehensive battery of test to evaluate different cognitive abilities in the participants and trying to correlate with plasma levels of different diet antioxidants, and with the physical performance analysed by different scores. The weakness consists in the low number of participants that contribute to obtain correlation with few cognitive test (but despite the low sample number, some correlations are obtained). An additional aspect has to do with the analysis of the obtained data that could probably be processed more to try to find additional correlations; not with the raw data, but by groups of patients within certain ranges in each parameter analyzed; or with the three intervention groups with respect to physical exercise, for example.

We thank the reviewer for his/her positive feedback. Our answers are enclosed below.

Authors indicate that participants were randomly allocated in three different interventions regarding to physical activity. Author should group these participants and check correlations with LAPAQ or TUG values, with cognitive tests results (and with MCI), and even regarding to other parameters analysed.

Thank you. As explained in the third paragraph on page 3, the present analyses refer to the baseline database of study participants displaying all investigated parameters for the purpose of the study (i.e., physical activity measurements, cognitive tests, micronutrient levels and information on fruit/vegetable intake). No exercise intervention was applied to the study population at that point.

Author should discuss the reason why the correlation between one lipophilic micronutrient is only with one physical activity parameter or cognitive test parameter and not with the other ones. Is there correlation between the different physical activity parameters analysed? or is there correlation between the different cognitive test analysed? If there is not correlation, author should discuss the reason. If there is correlation, author should group all these activity parameters or all these cognitive test results and correlate them with the micronutrients levels. Additionally, is there correlation between fruit and vegetables intake and micronutrients levels? Authors should comment these correlations indicating even if they do not correlate with a brief discussion about why do not correlate.

We thank the reviewer for this important point which enabled us to show additional correlations which most of researchers in the field consider implicit but are indeed relevant for discussion and research outlooks. The table below reports the correlations found and has been added and discussed in the paper now.

MoCA

TMT A

TMT B

LaPAQ

Steps

TUG

LF

CF

CogState

MoCA

-0.6

<0.0001

-0.5

<0.0001

0.4

0.003

-0.3

0.005

0.35

0.001

0.5

<0.0001

ISLT: 0.6

<0.0001

TMT A

0.7

<0.0001

-0.3

0.03

0.3

0.02

-0.24

0.03

-0.46

<0.0001

IDNlog: 0.46

<0.0001

OCL: -0.3

0.009

ONB: -0.3

0.01

ISLT: -0.4

0.001

TMT B

0.4

<0.0001

-0.34

0.002

-0.45

<0.0001

DETlog:

IDNlog: -0.42

<0.0001

OCL: -0.37

0.001

ONB: -0.38

0.001

ISLT: -0.38

0.001

LaPAQ

Steps

0.34

0.009

ISLT:0.26

<0.05

TUG

-0.34

0.004

ISLT: -0.3

0.009

LF

0.48

<0.0001

ISLT: 0.3

0.009

CF

0.34

0.009

ISLT: 0.5

<0.0001

CogState

Table 1 should appear in only one page (authors can make it smaller if it not fit). Some legend would be necessary to explain gender parameter and an explanation how to obtain BMI. There are some parameters that present a great error… could it be better to show SEM instead of SD? In the case of parameters with big error, some descriptive analysis of the parameter would be needed (how is the distribution? It is possible to make different groups? And do they correlate with micronutrients levels?). Some description from data obtained is needed.

Thank you very much.  The tables will be arranged so that they fit in one page. Gender and BMI legends have been added. Although we agree the large standard deviations for micronutrients are somehow optically unpleasant, we decided over the years to avoid SE because the large SD represent in fact the physiology of circulating carotenoids, whose plasma concentration large ranges in the population are known and due to differences in dietary supplementation as well as to endogenous factors influencing uptake, distribution, metabolism and excretion (Bohn et al., 2017). For the same reasons the population size in our study is not powered to disclose correlations between each of the 8 compounds measured and the large variety of fruits and vegetables available for consumption. In any case, and according to the reviewer’s suggestions, we were able to identify two groups of individuals based upon micronutrient levels. The first group (7 subjects) was called “supermicro” as persons displayed all micronutrient levels above the median value (the Boolean AND operation was used in the selection algorithm).  These were compared to the group of the remaining 49 participants. This comparison undercovered the supermicro group as indeed better both physically (Timed Up&Go test, 8.0±0.9 sec vs. 9.2±1.4 sec, p = 0.002) and cognitively (One card learning subtest of the CogState, 58.6 ± 8.3 vs 65.7 ± 8.4, p = 0.04) performing than the rest of the participants; however, larger-sized studies are needed to adequately address the representativity of this effect in the general population.

What is the meaning of arguing about the levels of lutein and zeaxanthin in macula? This part of the discussion could be eliminated. It would be preferable to broaden the discussion regarding the cognitive functions that micronutrients levels correlate mainly and why other cognitive functions do not correlate.

The macula data are indeed in the context of their role as component of nutritional cognitive neuroscience, sharing common pathophysiological mechanisms of neurodegeneration. However, and in light of the further data analysis, we added in the discussion some text regarding observed correlations.

A more complet analysis of data should be needed do to the low number of participants included.

We thank the reviewer for his/her comments; we added more interesting analyses in the paper.

Reviewer 2 Report

This study shows an association between some nutritional biomarkers in the plasma to scores in physical and cognitive tests in a sample of mild cognitively impaired elderly individuals.

Table 1 contains the scores of some physical/cognitive tests along with the plasma level of some micronutrients. These have been derived from a sample of elderly cognitive individuals. The absence of a ‘control’ group of non-impaired individuals (even a historical control) makes it difficult to have an idea of exactly how much these values vary relatively.

Lines 259-60: The reference (43) is not in the numerical order, the preceding references being (37-39). The authors may have meant this to be (37) instead, which actually mentions the difference in the lutein concentration in the serum and brain.

Line 261: The reference cited here for lutein concentrations in the macula and brain is a study in rhesus monkeys. It may be advisable to mention this fact.

Line 231: ‘…and inversely associated to cognitive performance an biomarkers…’. Besides grammar, the authors may want to change the statement to “positively associated with cognitive performance and inversely associated with markers of oxidative stress.”

Minor points:

Line 85: ‘..as in possess of..’. Please check.

Line 246: Carotenods.

Author Response

Reviewer 2

This study shows an association between some nutritional biomarkers in the plasma to scores in physical and cognitive tests in a sample of mild cognitively impaired elderly individuals.

Table 1 contains the scores of some physical/cognitive tests along with the plasma level of some micronutrients. These have been derived from a sample of elderly cognitive individuals. The absence of a ‘control’ group of non-impaired individuals (even a historical control) makes it difficult to have an idea of exactly how much these values vary relatively.

As commented at the beginning of the second half of the last page as well as described in the results (citations 27 and 28), our data on carotenoid levels are consistent with those described in the literature from us and other groups. Most recently, in a survey study including a healthy European population (Donovan et al 2017), average values for different carotenoids were: lutein 0.29 µM, zeaxanthin 0.06 µM, ß-cryptoxanthin 0.29 µM, lycopene 0.65 µM, a-carotene 0.21 µM, and ß-carotene 0.66 µM. Similar ranges of carotenoids were reported in a cross sectional study in six European countries (Stuetz et al 2016). The vitamin levels reported are comparable to those of our study with a-tocopherol 27.8 µM, g-tocopherol 1.3 µM, and retinol 1.7 µM. It should be noted, however, that micronutrient levels in populations vary over a great range due to differences in dietary supple and endogenous factors influencing uptake, distribution, metabolism and excretion (Bohn et al 2017).

Lines 259-60: The reference (43) is not in the numerical order, the preceding references being (37-39). The authors may have meant this to be (37) instead, which actually mentions the difference in the lutein concentration in the serum and brain.

Thank you, corrected.

Line 261: The reference cited here for lutein concentrations in the macula and brain is a study in rhesus monkeys. It may be advisable to mention this fact.

Thank you, implemented.

Line 231: ‘…and inversely associated to cognitive performance an biomarkers…’. Besides grammar, the authors may want to change the statement to “positively associated with cognitive performance and inversely associated with markers of oxidative stress.”

Thank you, corrected.

Minor points:

Line 85: ‘..as in possess of..’. Please check.

Thank you, corrected.

Line 246: Carotenods.

Thank you, corrected.

Round 2

Reviewer 1 Report

Authors have addressed most suggestions and argued the ones not addressed.